# Histone Deacetylase 6 Inhibitor CKD-WID Suppressed Monosodium Urate-Induced Osteoclast Formation by Blocking Calcineurin-NFAT Pathway in RAW 264.7 Cells

**DOI:** 10.3390/ph16030446

**Published:** 2023-03-16

**Authors:** Seong-Kyu Kim, Jung-Yoon Choe, Ji-Won Kim, Ki-Yeun Park

**Affiliations:** 1Division of Rheumatology, Department of Internal Medicine, Catholic University of Daegu School of Medicine, Daegu 42472, Republic of Korea; 2Arthritis and Autoimmunity Research Center, Catholic University of Daegu, 33, Duryugongwon-ro 17-gil, Nam-gu, Daegu 42472, Republic of Korea

**Keywords:** histone deacetylase, CKD-WID, osteoclast, monosodium urate, calcineurin

## Abstract

Histone deacetylase (HDAC) has been found to play a crucial role in the regulation of osteoclast differentiation and formation. This study was designed to identify the effect of the HDAC6 inhibitor CKD-WID on the receptor for the activation of nuclear factor-κB ligand (RANKL)-mediated osteoclast formation in the presence of monosodium urate (MSU) in RAW 264.7 murine macrophage cells. The expression of osteoclast-specific target genes, calcineurin, and nuclear factor of activated T-cells cytoplasmic 1 (NFATc1) was evaluated in RAW 264.7 murine macrophages treated with MSU, RANKL, or CKD-WID by real-time quantitative polymerase chain reaction and Western blot assay. The effect of CKD-WID on osteoclast formation was measured by tartrate-resistant acid phosphatase (TRAP) staining, F-actin ring formation staining, and assays for bone resorption activity. RANKL in the presence of MSU significantly induced HDAC6 gene and protein expression in RAW 264.7 cells. CKD-WID markedly suppressed the expression of osteoclast-related markers such as c-Fos, TRAP, cathepsin K, and carbonic anhydrase II induced by co-stimulation with RANKL and MSU in RAW 264.7 cells. Transcription factor NFATc1 mRNA expression and nuclear NFATc1 protein expression induced by co-stimulation with RANKL and MSU were significantly inhibited by CKD-WID treatment. CKD-WID also decreased the number of TRAP-positive multinuclear cells and F-actin ring-positive cells and attenuated bone resorption activity. Co-stimulation with RANKL and MSU increased calcineurin gene and protein expression, which was significantly blocked by CKD-WID treatment. The HDAC6 inhibitor CKD-WID suppressed MSU-induced osteoclast formation through blocking the calcineurin-NFAT pathway in RAW 264.7 cells. This suggests that HDAC6 is considered a therapeutic target in uric acid-mediated osteoclastogenesis.

## 1. Introduction

Gout is a chronic inflammatory arthritis characterized by excessive accumulation of uric acid at intraarticular and periarticular structures [1]. Many studies have elucidated the cellular and molecular pathways involved in bone erosion in patients with chronic gout [1,2,3,4]. Osteoclast-like multinucleated cells were identified within tophi and at the interface between soft tissue and bone by quantitative immunohistochemical analysis in gouty arthritis [2,3]. Dysregulated osteoclast formation has been observed in patients with tophaceous gout with bone erosion in the analysis of peripheral blood mononuclear cells and bone tissues [2]. Activation of the receptor that activate the nuclear factor-κB (RANK) and RANK ligand (RANKL) pathways in the presence of monosodium urate (MSU) crystal deposition leads to development of bone erosions in gouty arthritis [2,4]. Recently, a fully human monoclonal antibody directed against RANKL, denosumab, has been developed as a therapeutic option for bone loss in diverse pathological conditions such as metastatic lytic bone lesions, bone erosion in inflammatory arthritis, or low bone mineral density in osteoporosis [5]. Despite advances in the treatment of bone loss diseases, efforts on the identification of potent therapeutic agents with anti-osteoclast effect in gouty arthritis are still ongoing. 

Acetylation at lysine residues in histone proteins plays a crucial role in gene expression in eukaryotic cells. Histone deacetylases (HDACs) are enzymes that remove acetyl groups from histone and regulate the transcriptional process, acting as transcriptional co-repressors that reduce target gene expression [6]. HDACs are involved in the pathogenesis of diverse inflammatory and hematological diseases. In addition, these enzymes have been identified to be involved in bone development and skeletal diseases [7,8]. Eighteen HDAC genes have been identified in the human genome and are divided into classes I, II, III, and IV. Some HDAC enzymes such as HDAC3, HDAC7, and SIRT1 have been demonstrated to be related to regulation of osteoclast differentiation and formation [7]. Several HDAC inhibitors such as panobinostat, ACY-1215, FR901228, and trichostatin A suppress osteoclast differentiation and formation through downregulation of diverse pro-osteoclastogenic molecules [9,10,11,12]. In addition, clinical trials of selective inhibitors for HDAC6, a class II HDAC, also exhibited therapeutic effects in diverse lymphoproliferative and hematological diseases [13]. Considering the association between HDAC6 and bone lesions, some investigations on HDAC6 related to osteoclastogenesis have been performed. Two studies demonstrated that HDAC6 did not play a meaningful role in osteoclast differentiation and formation [14,15]. In contrast, functional loss of HDAC6 led to efficiently downregulated osteoclast differentiation and osteoclastogenic potency in different experimental cell lines [10,12,16]. The role of HDAC6 in osteoclasts differentiation and formation remains to be clarified in other pathogenic conditions.

Studies have shown that HDAC6 is involved in the activation of the NACHT, LRR, and PYD-domains-containing the protein 3 (NLRP3) inflammasome, playing an important role in the pathological mechanism of gout [17]. In addition, HDAC1/2 dual inhibition was shown to be an effective strategy in treating gout [18]. The role of HDAC6 in the regulation of osteoclasts in bone damage in the pathogenesis of gout is not fully determined. HDAC inhibition has been found to be related to the suppression of osteoclast formation through regulation of the calcineurin-nuclear factor of activated T-cells cytoplasmic 1 (NFATc1) pathway [9,10,11,12]. Among HDAC6 inhibitors, the mechanism by which CKD-WID regulates osteoclast formation and differentiation in various pathological diseases has not yet been elucidated. Furthermore, the HDAC6 inhibitor CKD-WID has not been well studied regarding its therapeutic role in MSU crystal-mediated bone damage in the pathogenesis of gout. The main objective of this study was to examine the effect of CKD-WID on the calcineurin-NFATc1 pathway in MSU-induced osteoclast formation.

## 2. Results

### 2.1. RANKL and MSU Induce HDAC6 Expression in RAW 264.7 Cells

We first assessed the expression of the HDAC6 mRNA and protein in RAW 264.7 cells in response to RANKL and/or MSU. RAW 264.7 cells stimulated with either MSU or RANKL showed approximately two-fold and four-fold increases in HDAC6 mRNA expression compared with non-stimulated cells (Figure 1A). HDAC6 mRNA expression was also significantly increased in cells stimulated with both MSU and RANKL compared with non-stimulated cells. Consistently, Western blot and densitometric analyses showed that either RANKL or MSU increased HDAC6 protein level compared with that in non-stimulated cells (Figure 1B). Co-stimulation of both MSU and RANKL markedly increased HDAC6 protein expression. The HDAC6 inhibitor CKD-WID dose-dependently suppressed HDAC6 mRNA and protein expression under stimulation with both RANKL and MSU compared with levels in RAW 264.7 cells treated with only RANKL and MSU (Figure 1C,D). These results suggest that HDAC6 is involved in RANKL-induced osteoclast formation in the presence of MSU.

### 2.2. CKD-WID Regulates Osteoclast-Related Factors Induced by RANKL and MSU Stimulation

RAW 264.7 cells stimulated with both MSU and RANKL showed significantly increased mRNA expression of the osteoclast transcription factors c-Fos, TRAP, cathepsin K, and carbonic anhydrase II (Figure 2A). Treatment with CKD-WID (1.0 or 3.0 μM) significantly attenuated c-Fos, TRAP, cathepsin K, and carbonic anhydrase II mRNA expression. 

Next, we assessed whether CKD-WID suppressed the osteoclast-related markers c-Fos, TRAP, cathepsin K, and carbonic anhydrase II protein expression in RAW 264.7 cells treated with both MSU crystal and RANKL. Consistent with gene expression, Western blot assay and densitometry revealed that CKD-WID treatment at 0.5, 1.0, or 3.0 μM inhibited protein expression of these osteoclast-related markers (Figure 2B).

RAW 264.7 cells treated with CKD-WID showed increased IRF-8 mRNA and protein levels compared with levels in cells treated with only MSU and RANKL (Figure 2C,D). We also found that the Blimp1 gene and protein, which are induced by NFATc1, were inhibited in cells treated with CKD-WID compared with cells treated with only MSU and RANKL alone. 

### 2.3. CKD-WID Suppresses Osteoclast Formation Induced by RANKL and MSU

TRAP staining analysis was performed to assess the effect of CKD-WID on multinucleated osteoclast formation in RAW 264.7 cells treated with both RANKL and MSU. In addition, we evaluated the formation of TRAP-positive cells by stimulation with either RANKL or MSU crystals. The number of TRAP-positive multinucleated cells under stimulation with both RANKL and MSU was much higher than those stimulated with either RANKL or MSU crystals and markedly decreased in the cells treated with CKD-WID at doses of 0.5 and 1.0 μM, compared with cells without CKD-WID (Figure 3A). In particular, the number of TRAP-positive cells was significantly reduced with a higher dose of CKD-WID (3.0 μM). We calculated the fusion index. Fusion index under stimulation with both RANKL and MSU crystals was higher than when stimulated with either RANKL or MSU crystals, and it was significantly decreased in cells treated with CKD-WID, compared to those cultured with both MSU and RANKL alone without CKD-WID (Figure 3B). Actin ring formation at the periphery of mature osteoclasts cultured with both RANKL and MSU was markedly reduced in cells treated with CKD-WID in a dose-dependent manner (Figure 3C). Bone resorption assay revealed that the area of bone resorption was significantly reduced in RAW 264.7 cells stimulated with both MSU and RANKL upon treatment with CKD-WID in a dose-dependent manner (Figure 3D). In addition, CKD-WID at doses of 1.0 and 3.0 μM under stimulation with both RANKL and MSU markedly attenuated bone resorption activity, compared with cells without CKD-WID treatment (Figure 3E).

### 2.4. CKD-WID Inhibits Calcineurin-NFATc1 Pathway during Osteoclast Formation

Calcineurin-NFATc1 signaling is a crucial mechanism in osteoclast differentiation [19]. We assessed whether CKD-WID affected calcineurin and NFATc1 expression in RANKL-induced osteoclast formation in the presence of MSU. Calcineurin and NFATc1 mRNA expression were significantly induced by co-stimulation of both MSU and RANKL (Figure 4A). Treatment with CKD-WID (1.0 or 3.0 μM) induced a decrease in calcineurin and NFATc1 mRNA expression. Consistently, Western blot quantification by densitometry showed that CKD-WID at dosages of 1.0 or 3.0 μM inhibited calcineurin expression in RAW 264.7 cells treated with both MSU and RANKL (Figure 4B). CKD-WID also suppressed the protein expression of nuclear rather than cytoplasmic NFATc1, indicating that CKD-WID inhibited translocation of NFATc1 to the nucleus. We investigated whether the HDAC6 inhibitor CKD-WID affected functional activity of calcineurin under stimulation of MSU and RANKL in the assessment of the relationship between HDAC6 and calcineurin (Figure 4C). We found that both MSU and RANKL markedly induced calcineurin ubiquitination, which was gradually inhibited by CKD-WID in a dose-dependent manner. It might suggest that CKD-WID accelerates the ubiquitin–proteasome degradation of calcineurin.

Higher mRNA expression of osteoclast-related markers c-Fos, NFATc1, TRAP, cathepsin K, and carbonic anhydrase II in RAW 264.7 cells treated with both RANKL and MSU were significantly suppressed by CKD-WID treatment, similarly with the anti-osteoclastic effect of the calcineurin inhibitors cyclosporin A or FK506 (Figure 4D). Western blot and densitometry showed that levels of osteoclast-related markers c-Fos, NFATc1, and cathepsin K protein expression were attenuated by cyclosporin A, FK506, and CKD-WID (Figure 4E). However, only CKD-WID suppressed TRAP and carbonic anhydrase II protein expression under stimulation with MSU crystals and RANKL. 

### 2.5. HDAC6 Knockdown Blocks Osteoclast Formation through Calcineurin-NFATc1 Pathway

HDAC6 siRNA was used to block HDAC6 gene expression to verify that HDAC6 is involved in RANKL-induced osteoclast formation via the calcineurin-NFATc1 pathway. Figure 5A shows that RAW 264.7 cells transfected with HDAC6 siRNA attenuated mRNA expression of c-Fos, TRAP, cathepsin K, and NFATc1 and calcineurin, compared to non-transfected cells. Consistently, Western blot and densitometric analyses showed that protein expression of c-Fos, TRAP, cathepsin K, and NFATc1 and calcineurin in HDAC6 knockdown was significantly inhibited compared to non-transfected cells (Figure 5B). 

## 3. Discussion

Joint damage accompanied by bone erosion and cartilage destruction is an important clinical feature of chronic tophaceous gout, as observed with other inflammatory joint diseases including rheumatoid arthritis or psoriatic arthritis [1,20]. Osteoclasts are an active key player in the development of bone erosion in gout. Activation of the transcription factors nuclear factor-κB (NF-κB), NFATc1, and calcineurin is required for initiation of the RANK-RANKL pathway in osteoclastogenesis [19]. Research has shown that osteoclastogenesis in gout is tightly regulated by the RANK-RANKL mechanism, which is similar to the process of osteoclast formation in other forms of inflammatory arthritis [2,21,22]. The possibility that HDACs are involved in the osteoclast formation in the pathogenesis of gout has been proposed. HDAC6 inhibition suppressed NLRP3 inflammasome activation by attenuating the transcription of NF-κB in the priming process and by reducing the expression of mature caspase-1, ultimately resulting in reduced production of IL-1β [17]. Cleophas et al. showed that the HDAC1/2 inhibitor Romidepsin potently decreased the production of numerous inflammatory cytokines including IL-1β, IL1Ra, IL-6, and IL-8 in PBMCs incubated with MSU and palmitic acid, suggesting HDAC1/2 inhibition as a novel therapeutic strategy in acute gout [18]. Whether HDAC6 is involved in or responsible for MSU-induced bone damage in gout had not been examined. The purpose of this study was to identify whether HDAC6 inhibition could attenuate osteoclast differentiation and formation through the RANK-RANKL-NFATc1 pathway in gout. The main findings of this study are that the selective HDAC6 inhibitor CKD-WID inhibited the formation of osteoclasts differentiated from RANKL-primed RAW 264.7 murine macrophages treated with MSU crystals. In particular, we observed that CKD-WID regulated osteoclast formation through suppressing calcineurin in gout. 

Eighteen mammalian HDAC members have been found to be involved in bone development and skeletal diseases through functional regulation of osteoblast, osteoclast, and chondrocyte differentiation and formation. Knockdown of HDACs such as HDAC2 and HDAC3 resulted in the downregulation of osteoclast-related genes such as *Nfatc1* and *Dc-stamp* and suppression of osteoclast activity, whereas inhibition of HDAC4, HDAC7, and SIRT1 increased osteoclast differentiation and accelerated TRAP-positive multinucleated osteoclast formation through the upregulation of osteoclast-related genes including *Nfatc1*, *c-Fos*, *Ctsk*, and *Dc-stamp* [7,8]. Several studies have been conducted on the potential role of HDAC6 in osteoclast differentiation and formation. HDAC6 mRNA expression was not significantly increased in mouse bone marrow-deprived macrophages (BMMs) cultured with M-CSF and RANKL [14]. In addition, there was no significant change in HDAC6 gene expression during a time course analysis of osteoclastogenesis. Similarly, HDAC6 knockdown using HDAC6 siRNA in mouse BMMs did not influence the expression of osteoclast-specific genes including *c-Fos*, *Nfatc1*, *Ctsk*, and *Dc-stamp*, and HDAC6 inhibition did not induce a significant change in number or size of TRAP-positive multinuclear cells [15]. In contrast, Wang et al. demonstrated that HDAC6 inhibition in dental mesenchymal stem cells increased mineralized nodule formation and HDAC6 knockdown using siRNA-inhibited mRNA expression of osteoclast markers *TRAP* and *Ctsk*, resulting in osteoclast differentiation [16]. A selective HDAC6 inhibitor ACY-1215 alone and in combination with bortezomib suppressed osteoclast differentiation from PBMCs stimulated with RANKL and resulted in a decreased number of TRAP-positive multinucleated cells and bone-resorbing potency [12]. Another HDAC6 inhibitor panobinostat, which is used as an anticancer drug for multiple myeloma (MM), combined with FK506 significantly affected osteoclast formation through the reduction of PPP3CA expression [10]. Consistently, our results also confirmed that HDAC6 inhibition using the selective HDAC6 inhibitor CKD-WID suppressed osteoclast differentiation and formation by inhibition of osteoclastogenic genes such as c-Fos, TRAP, cathepsin K, and carbonic anhydrase II in MSU-primed RAW 264.7 cells in the presence of RANKL. Understanding the cellular and molecular mechanisms underlying the effect of HDAC6 on osteoclast differentiation and formation might offer a potent therapeutic opportunity to manage osteoclast-related bone diseases.

In this study, the effect of a selective HDAC6 inhibitor CKD-WID on the signal transduction of RANKL-induced osteoclast differentiation under stimulation with MSU crystals in macrophages was analyzed. Activation of downstream signaling molecules in the RANK-RANKL system is responsible for osteoclast differentiation and formation [19]. Multiple mechanisms by which HDAC inhibitors modulate osteoclastogenesis through the regulation of these molecules have been identified. Nakamura et al. showed that the HDAC inhibitor FR901228 blocked both RANKL-induced NFATc1 activation and increased mRNA expression of IFN-β in rat bone marrow cells treated with RANKL, resulting in the inhibition of osteoclastogenesis and a significant reduction in bone erosion in a Complete Freund’s Adjuvant–induced rat model [11]. IFN-β neutralizing antibody markedly increased the number of TRAP-positive multinuclear cells by reversing the inhibition of osteoclastogenesis induced by FR901228. Enhanced expression of C/EBP-b and MKP-1 by trichostatin A together with the downregulation of c-Fos and NFATc1 in RAW 264.7 cells was found to be related with inhibition of osteoclastogenesis [9]. Among transcription factors and osteoclast-specific markers that are involved in the activation of the calcineurin-NFAT pathway after binding to RANK and RANKL, calcineurin, a serine/threonine phosphatase, is activated by the binding of calcium-calmodulin [23] and induces dephosphorylation of NFAT transcription factors in the cytoplasm, facilitating their nuclear translocation to induce osteoclast-specific target genes involved in osteoclast formation [24]. Imati et al. demonstrated that the anti-osteoclastic effect of the HDAC6 inhibitor panobinostat was related with the degradation of PPP3CA, a catalytic subunit of calcineurin, in MM cell lines [10]. The anti-osteoclastic effect of the HDAC6 inhibitor ACY-1215 was induced by blocking osteoclast-related transcription factors including p-ERK, p-AKT, c-Fos, and NFATc1 [12]. Consistently, this study confirmed that HDAC6 inhibition using CKD-WID also showed an anti-osteoclastic effect through the suppression of calcineurin and NFATc1 activation downstream in the RANK-RANKL pathway.

Activation of NFATc1, c-JUN, and TNF receptor-associated factor-6 (TRAF6) downstream of the RANK-RANKL signal pathway has also been regarded as a crucial step for formation in osteoclast-like cells in the pathogenic mechanism of gout [21,25]. Although anti-osteoclastogenic mechanisms of HDAC6 inhibition in the process of osteoclastogenesis in dental bone growth and MM have been identified [12,16], there are no data on the mechanism of HDAC6 inhibition involved in bone erosion in gout arthritis. This study confirmed that CKD-WID markedly attenuated the expression of pro-osteoclastogenic genes and proteins c-Fos, cathepsin K, TRAP, and carbonic anhydrase II; decreased the expression of the Blimp1 gene and protein, a transcriptional regressor of anti-osteoclastogenic factor such as IRF-8; and enhanced dephosphorylation of NFATc1 by calcineurin. Until now, there have been no data on calcineurin in bone erosion in gouty arthritis. An early study showed that panobinostat suppressed osteoclastogenesis through the downregulation of PPP3CA, a subunit of the calcineurin complex [10], as shown by the inhibitory effects of FK506 on PPP3CA and the calcineurin complex [26]. We also found that CKD-WID enhanced the degradation of calcineurin in RANKL-stimulated RAW 264.7 cells. MSU-primed RAW 264.7 cells in the presence of RANKL showed significantly induced expression of calcineurin mRNA and protein, which was inhibited by CKD-WID in a dose-dependent manner. The present study confirmed that the inhibitory effect of CKD-WID on calcineurin is consistent with FK506 and cyclosporin A in blocking translocation of NFAT to the nucleus, which was also identified in another study through the inhibition of calcineurin phosphatase activity [27]. These results suggest that the HDAC6 inhibitor has a therapeutic effect on calcineurin-targeted osteolytic lesions, similar to the effects of FK506 and cyclosporin A, with inhibitory activity against calcineurin and the NFAT transcription factor. This study did not verify the osteoclast inhibitory ability with calcineurin inhibitors FK506 and cyclosporin A, compared to CKD-WID. Thus, it is necessary to identify the difference in therapeutic efficacy of CKD-WID with FK506 and cyclosporin A in future studies.

## 4. Materials and Methods

### 4.1. Cell Culture and Osteoclast Differentiation

RAW 264.7 cells were purchased from the Korean Cell Line Bank (KCLB, Seoul, Republic of Korea) and maintained in α-minimal essential medium (α-MEM) (Gibco, BRL, Grand Island, NY, USA) supplemented with 10% fetal bovine serum (Hyclone, Logan, UT, USA), 100 U/mL penicillin, and 100 μg/mL streptomycin. To differentiate RAW 264.7 cells into osteoclasts, cells were plated in 24-well plates (1 × 10^4^ cells/well) with α-MEM containing sRANKL (100 ng/mL) and MSU crystals (0.1 mg/mL) and cultured for 8 days.

### 4.2. Material Reagents and Antibodies

Soluble recombinant mouse RANKL (sRANKL) was purchased from Peprotech (Rocky Hill, NJ, USA). The primary antibodies used were as follows: anti-TRAP, anti-cathepsin K, anti-c-Fos, anti-NFATc1, anti-Blimp-1, and anti-β-actin from Santa Cruz Biotechnology (Santa Cruz, CA, USA); anti-carbonic anhydrase II, anti-cathepsin K, anti-HDAC6, and anti-IRF-8 from Abcam (Cambridge, MA, USA); and anti-Calcineurin from by MyBiosource (San Diego, CA, USA). Calcineurin inhibitors FK506 and cyclosporin A were purchased from Sigma-Aldrich (Saint Louis, MO, USA) and Abcam (Cambridge, MA, USA), respectively; the inhibitors were dissolved in dimethyl sulfoxide (DMSO, Sigma-Aldrich) and stored at −20 °C. HDAC6 inhibitor CKD-WID was kindly provided by the Chong Kun Dang Pharmaceutical Corp. (Seoul, Korea). 

### 4.3. Quantitative Real Time-Polymerase Chain Reaction (RT-PCR) 

Cells were seeded in 24-well plates at 1 × 10^4^ cells and treated with sRANKL (100 ng/mL) and MSU crystals (0.1 mg/mL) for 4 days. Various concentrations of CKD-WID (0.5, 1.0, and 3.0 µM) were added to the medium, and cells were further cultured for 4 days. Total RNA of cells was extracted with TRIzol Reagent (Gibco BRL, Grand Island, NY, USA). Complementary DNA was synthesized from 1 μg of RNA using a ReverTra Ace-α- reverse transcriptase kit (Toyobo, Osaka, Japan) in a reaction condition of incubation at 37 °C for 15 min, 50 °C for 5 min, and 98 °C for 5 min.

RT-PCR was performed using the Mini Option TM Real-time PCR system (Bio-Rad, Hercules, CA, USA) with SYBR Green PCR Master Mix (Toyobo, Osaka, Japan) according to the manufacturers’ instructions. The PCR amplification reaction (total volume of 20 µL) contained 2 µL of cDNA, 10 µL of SYBR^®^ Green Realtime PCR Master Mix, 10 pmol/L of each primer, and 6.4 µL of distilled water. The reactions were carried out with an initial denaturation at 95 °C for 15 min, followed by 40 cycles of 95 °C for 5 s, 55–62 °C for 30 s, and 72 °C for 15 s. All reactions were run in triplicate. 

### 4.4. Western Blot Assay 

Cells (2 × 10^5^) were seeded in 60 mm dishes and cultured with sRANKL (100 ng/mL) and MSU crystals (0.1 mg/mL) for 4 days, followed by treatment with cyclosporin A, FK506, or CKD-WID for 4 days. Cells were lysed using radio immunoprecipitation assay (RIPA) buffer (Pierce, Thermo Scientific, Waltham, MA, USA) containing a protease inhibitor cocktail 1 tablet (Roche Diagnostics, Mannheim, Germany) on ice for 10 min and centrifuged at 13,000 rpm for 10 min. The supernatant was collected and stored at −80 °C. Cytosolic and nuclear fractions were separated using the NE-PER™ nuclear and cytoplasmic extraction reagents (Thermo Scientific, Rockford, IL, USA) supplemented with Halt Protease and Phosphatase Inhibitor Cocktail (Thermo Scientific) in accordance with the manufacturer’s instructions. The samples were mixed with 6× Laemmli Buffer (Bio-Rad, Hercules, CA, USA) and denatured by boiling at 95 °C for 5 min, followed by separation by 10% SDS-PAGE gel electrophoresis and transfer to a nitrocellulose membrane (Bio-Rad). The membrane was probed with appropriate antibodies.

### 4.5. Immunoprecipitation Analysis

For immunoprecipitation, cells were lysed in NP-40 buffer and precleared with Protein A/G PLUS agarose beads (Santa Cruz, Dallas, TX, USA) for 2 h at 4 °C. The ubiquitination of calcineurin was immunoprecipitated using anti-calcineurin antibody for 24 h at 4 °C followed by incubation with Protein A/G PLUS agarose (Santa Cruz, CA, USA) for 3 h with rotation. The Beads were extensively washed 5 times with NP-40 buffer. The beads were eluted by boiling for 5 min at 95 °C and centrifuged for 5 min at 2500 rpm at 4 °C. The supernatant was collected, and the bound proteins were analyzed by Western blot analysis.

### 4.6. Transfection of HDAC6 siRNA

Cells (5 × 10^3^) were seeded in 24-well plates and transfected with mouse HDAC6 siRNA (MSS205078) and negative control siRNA at a final concentration of 50 nM using Lipofectamine RNAiMAX in Opti-MEM media (Gibco BRL, Grand Island, NY, USA) according to the manufacturer’s instructions. The mixtures were incubated for 10 min at room temperature for activation of transfection complexes. After culture for 48 h, transfected cells were collected and analyzed for RT-PCR and Western blot for target molecules.

### 4.7. Bone Resorption Activity

Bone resorption activity was evaluated using a bone resorption assay kit (COSMO Bio, Tokyo, Japan). Briefly, cells seeded on calcium phosphate-coated plates (1 × 10^4^ cells/well) were cultured with sRANKL (100 ng/mL) and MSU crystals (0.1 mg/mL) for 6 days and further treated with diverse concentrations of CKD-WID (0.5, 1, and 3 µM) for 4 days. Culture supernatants were placed into a black flat-bottom 96-well microplate, and resorption assay buffer was added. ELISA plate reader (BMG Lab Technologies, Offenburg, Germany) with 485 nm excitation and 520 nm emission was applied to measure the fluorescence intensity.

To examine bone resorption, the medium was removed from wells, and cells were added with 5% sodium hypochlorite for 5 min. After washing twice with PBS, wells were dried, and the resorption pits were photographed using a microscope (Olympus, Tokyo, Japan).

### 4.8. Tartrate-Resistant Acid Phosphatase (TRAP) Staining

At day 10 of differentiation, cells were fixed with 10% formalin solution for 5 min at room temperature and then stained using a TRAP staining kit (Takara Bio, Inc., Shiga, Japan) in accordance with the manufacturer’s instructions. Cells were washed with distilled water, and TRAP-positive multinucleated cells were observed by microscopy (Olympus, Tokyo, Japan). TRAP-positive multinucleated cells (≥3 nuclei) were counted using an inverted microscope (Olympus, Tokyo, Japan). 

### 4.9. Actin Ring Staining

Cells (2.5 × 10^4^/well in 6-well plates) were grown on glass coverslips and treated with sRANKL (100 ng/mL) and MSU crystals (0.1 mg/mL) for 6 days. Various concentrations of CKD-WID (0.5, 1, and 3 µM) were added to the medium, and cells were cultured for 4 days.

Cells were fixed with 4% paraformaldehyde for 10 min. After washing cells in PBS, F-actin rings were stained using Alexa Fluor™ 488 Phalloidin (Thermo Fisher Scientific, Waltham, MA, USA). Cells were stained with DAPI for 20 min, washed with PBS, and observed using fluorescence microscopy (TE2000-U, Nikon Instruments Inc., Melville, NY, USA).

### 4.10. Statistical Analysis

Data for continuous variables are described as mean ± standard error of the mean (SEM). The statistical differences in the expression of target genes between two groups were assessed by nonparametric Mann–Whitney U test. A *p* value 0.05 or less than was considered statistically significant. Statistical analysis was performed using GraphPad Prism Version 5.04 software (GraphPad Software, San Diego, CA, USA).

## 5. Conclusions

In conclusion, this study is the first to demonstrate that the HDAC6 inhibitor CKD-WID inhibits MSU-induced osteoclast differentiation and formation in RAW 264.7 cells (Figure 5C). CKD-WID attenuates osteoclast-specific genes and transcription factors such as NFATc1 in the process of osteoclastogenesis through inhibition of calcineurin responsible for dephosphorylation of cytoplasmic NFATc1. Furthermore, this is the first evidence of an interaction of gout-related bone damage and calcineurin. These data suggest calcineurin as a target for regulation of osteoclast formation in gout. Further studies are needed to verify the therapeutic effect of the HDAC6 inhibitor on the treatment of bone erosion in gouty arthritis.

## Figures and Tables

**Figure 1 pharmaceuticals-16-00446-f001:**
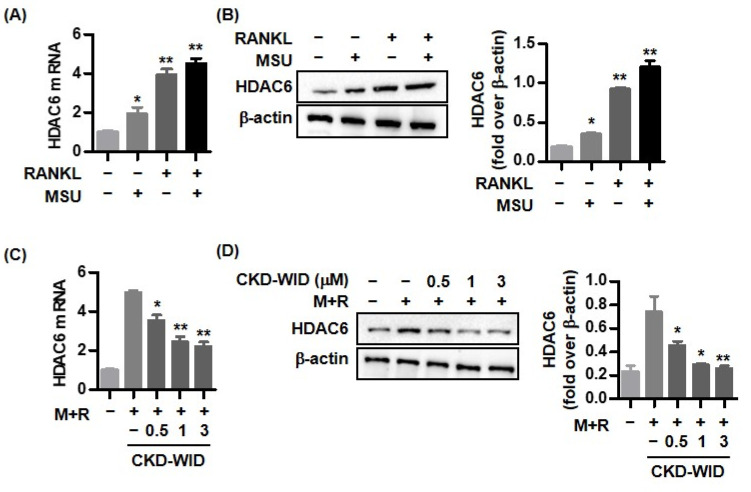
**Effect of both RANKL and MSU crystals on HDAC6 expression in RAW 264.7 cells**. (**A**,**B**) HDAC6 mRNA levels and protein bands were assessed in RAW 264.7 cells stimulated with MSU alone (0.1 mg/mL), RANKL alone (100 ng/mL), or both for 8 days. * *p* < 0.05 and ** *p* < 0.01 compared to non-stimulated cells. (**C**,**D**) RAW 264.7 cells by both MSU and RANKL stimulation for 4 days were followed by addition of different dosages of CKD-WID (0.5, 1.0, and 3.0 μM) for 4 days. HDAC6 mRNA and protein band expression were assessed by RT-PCR, Western blot assay, and densitometric analysis. * *p* < 0.05 and ** *p* < 0.01 compared to MSU and RANKL alone without CKD-WID treatment. Values presented as mean ± SEM of three independent experiments. Abbreviations: RANKL, receptor for activation of nuclear factor-κB ligand; MSU, monosodium urate; HDAC, histone deacetylase.

**Figure 2 pharmaceuticals-16-00446-f002:**
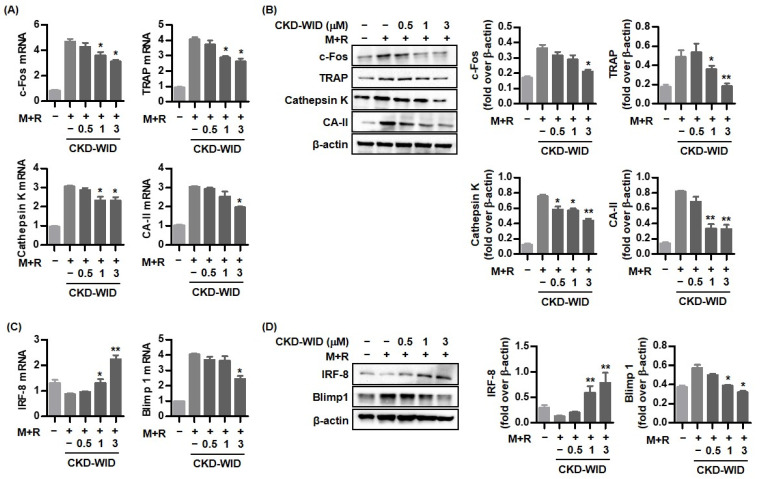
**Anti-osteoclastic effect of CKD-WID on molecules responsible for osteoclastogenesis.** (**A**,**B**) RANKL-induced RAW 264.7 cells in the presence of MSU were treated with three different dosages of CKD-WID (0.5, 1.0, and 3.0 μM) for 4 days to determine the mRNA and protein expression of osteoclast-related marker genes including c-Fos, TRAP, cathepsin K, and carbonic anhydrase II. * *p* < 0.05 and ** *p* < 0.01 compared to MSU and RANKL alone without CKD-WID treatment. (**C**,**D**) IRF-8 and Blimp 1 expression in RAW 246.7 cells precultured with both RANKL and MSU was measured after the addition of CKD-WID for 4 days. * *p* < 0.05 and ** *p* < 0.01 compared to MSU and RANKL alone without CKD-WID treatment. These results were evaluated by RT-PCR and Western blot quantification by densitometry. Values presented as mean ± SEM of three independent experiments. Abbreviations: M + R, monosodium urate and RANKL; TRAP, tartrate-resistant acid phosphatase; CA-II, carbonic anhydrase II.

**Figure 3 pharmaceuticals-16-00446-f003:**
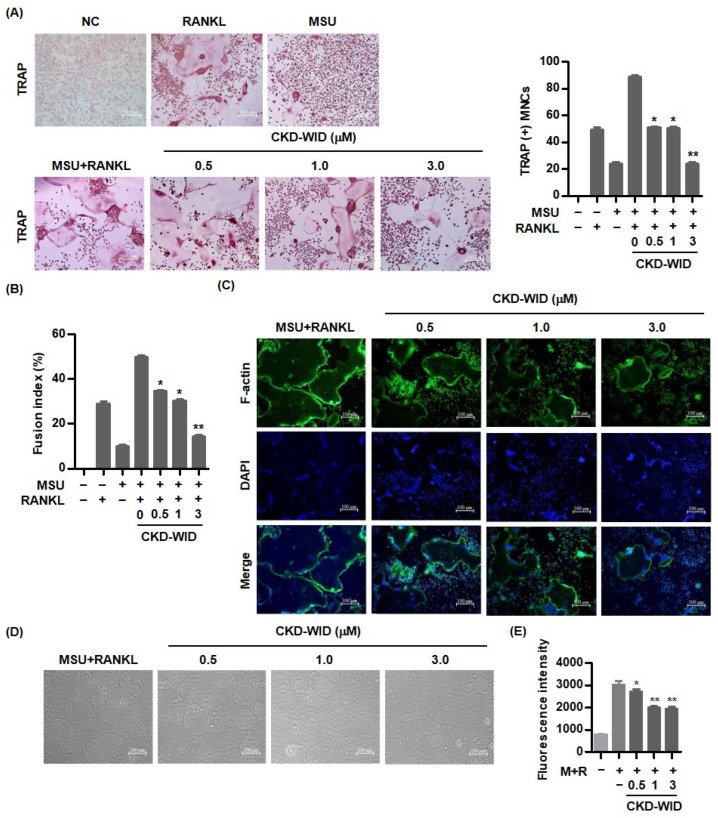
**CKD-WID suppresses osteoclast formation in RAW 264.7 cells treated with RANKL and MSU.** (**A**) CKD-WID (0.5, 1.0, and 3.0 μM) for 4 days was treated in RAW 264.7 cells differentiated into TRAP-positive multinuclear cells. Scale bar, 100 μm. * *p* < 0.01 and ** *p* < 0.001 compared to MSU and RANKL alone without CKD-WID treatment. (**B**) Fusion index was calculated to assess the degree of cell fusion by CKD-WID treatment. Fusion index (%) was determined by dividing the total number of nuclei in the giant cells by the total number of nuclei × 100. (**C**) Cells stimulated with RANKL and MSU for 6 days were fixed and stained for F-actin. The images of F-actin ring staining were illustrated after the addition of CKD-WID for 4 days. Scale bar, 100 μm. (**D**,**E**) Bone resorption assays and fluorescence intensity measurement were performed using osteoclasts differentiated from RAW 264.7 cells treated with RANKL and MSU for 6 days. Scale bar, 100 μm. * *p* < 0.01 and ** *p* < 0.001 compared to MSU and RANKL alone without CKD-WID treatment. Values presented as mean ± SEM of three independent experiments. Abbreviations: NC, negative control; RANKL, receptor for activation of nuclear factor-κB ligand; MSU, monosodium urate; TRAP, tartrate-resistant acid phosphatase; MNCs, multinucleated cells.

**Figure 4 pharmaceuticals-16-00446-f004:**
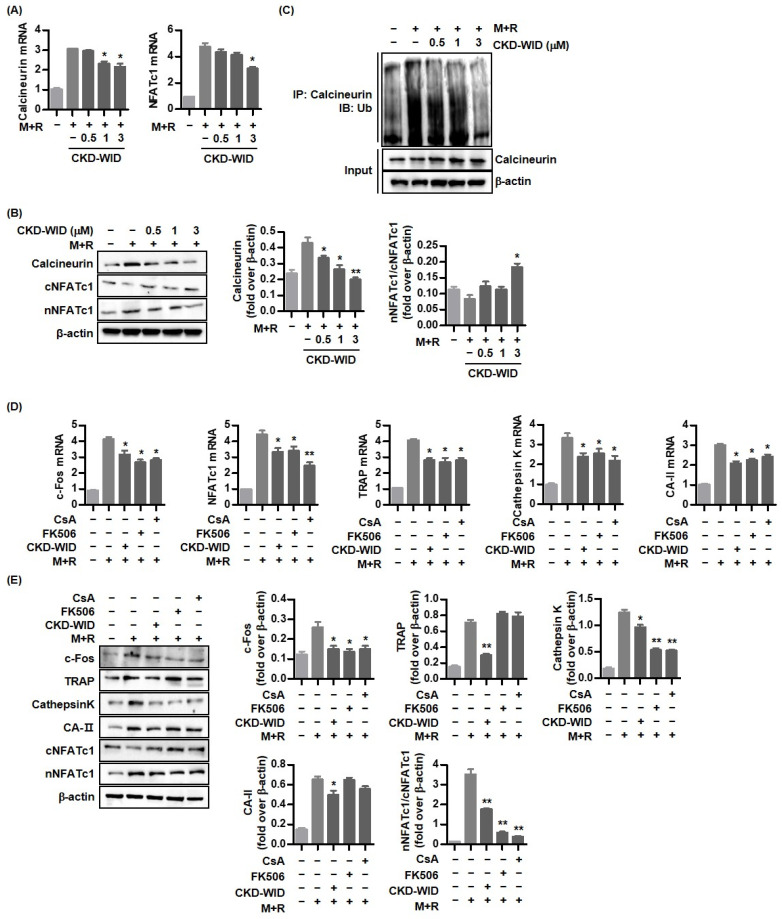
**CKD-WID inhibits RANKL-induced calcineurin-NFATc1 pathway in the presence of MSU.** (**A**,**B**) The calcineurin and NFATc1 mRNA and protein band expression were measured in RAW 264.7 cells differentiated into osteoclasts after the addition of CKD-WID for 4 days. * *p* < 0.05 compared to MSU and RANKL alone without CKD-WID treatment. (**C**) RAW 264.7 cells treated with both MSU and RANKL were cultured with CKD-WID. Ubiquitinated calcineurin in the cell extract was examined by immunoprecipitation and analyzed by Western immunoblot. (**D**,**E**) The mRNA and protein band expression for anti-osteoclastic effect of cyclosporin A, FK-506, and CKD-WID was evaluated. * *p* < 0.05 and ** *p* < 0.01 compared to MSU and RANKL alone without any kind of treatment. Abbreviations: M + R, monosodium urate and RANKL; NFATc1, nuclear factor of activated T-cells cytoplasmic 1; TRAP, tartrate-resistant acid phosphatase; CA-II, carbonic anhydrase II.

**Figure 5 pharmaceuticals-16-00446-f005:**
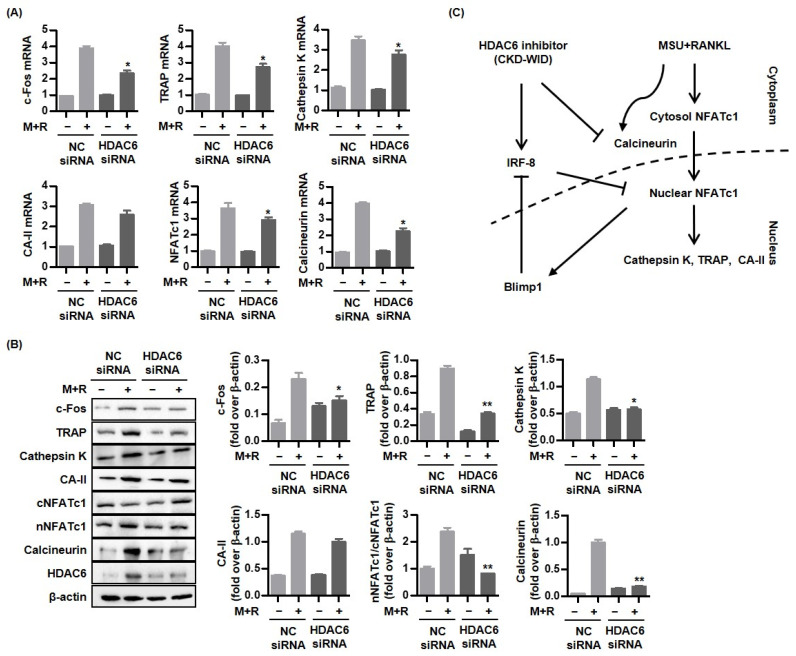
**HDAC6 knockdown promotes anti-osteoclastic activity through regulation of calcineurin-NFATc1 pathway.** (**A**,**B**) The mRNA and protein expression of osteoclatogenic molecules was measured in RAW 264.7 cells transfected with HDAC6 siRNA using RT-PCR and Western blot quantification by densitometry. * *p* < 0.05 and ** *p* < 0.01 compared to non-transfected cells. (**C**) Stimulation with both MSU and RANKL activates osteoclast-related markers c-Fos, NFATc1, calcineurin, cathepsin K, TRAP, and carbonic anhydrase II in RAW 264.7 cells. CKD-WID effectively suppresses osteoclast differentiation and formation through the downregulation of c-Fos, cathepsin K, TRAP, and carbonic anhydrase II in MSU crystal-induced inflammation. Blockage of calcineurin-NFAT pathway by CKD-WID leads to suppression of Blimp1, a transcriptional regressor of anti-osteoclastogenic factor such as IRF-8 in gouty inflammation. Abbreviations: M + R, monosodium urate and RANKL; NFATc1, nuclear factor of activated T-cells cytoplasmic 1; TRAP, tartrate-resistant acid phosphatase; CA-II, carbonic anhydrase II.

## Data Availability

The data underlying this article will be shared on reasonable request to the corresponding author.

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
