# Peer review of "Histone Deacetylase 6 Inhibitor CKD-WID Suppressed Monosodium Urate-Induced Osteoclast Formation by Blocking Calcineurin-NFAT Pathway in RAW 264.7 Cells"

_pharmaceuticals, 2023, doi:10.3390/ph16030446_

Round 1

Reviewer 1 Report

It is my opinion that the experiments and relative results provided are well organized and allow to support the initial hypothesis.

Some minor suggestions are listed below.

1. Line 50-52: This statememt is not correct as therapeutic agents have been already developed, for example denosumab, a monoclonal antibody against RANK. Please enanched this part by adding appropriate literature on the management of the chronic gouty arthritis

2. The paper has totally no literature on CKD-WID. It would be useful to describe CKD-WID and cite some studies using it. Moreover, it is not clear why CKD-WID was chosen over other specific HDAC6 inhibitors. Authors are recommended to enanched these points.

3. The discussion is quite confusing, difficult to read and appears to be overloaded in relation to the results obtained. Further, there is no a clear separation between results obtained and those from other studies. Moreover, a clear logic line of presentation is lacking. The result is that the scientific progress provided by this work do not reach the reader efficaciously. Thus, I recommend to review the writing of this chapter in particular I encourage to simplify the test by removing extra data not correlated with results (for example HSP90) and by better underlying the results obtained.

Author Response

Dear Editor

Manuscript ID: pharmaceuticals-2195802

Title: Histone deacetylase 6 inhibitor CKD-WID suppressed monosodium urate-induced osteoclast formation by blocking calcineurin-NFAT pathway in RAW 264.7 cells

   Thank for the editor and reviewers of the ‘Pharmaceuticals’ for reviewing our manuscript. We have made some corrections and clarifications in the revised manuscript according to the editor’s or reviewer's comments. You can find out tracing marks for changes in revised manuscript. The changes are summarized below:

It is my opinion that the experiments and relative results provided are well organized and allow to support the initial hypothesis. Some minor suggestions are listed below.

  1. Line 50-52: This statement is not correct as therapeutic agents have been already developed, for example denosumab, a monoclonal antibody against RANK. Please enanched this part by adding appropriate literature on the management of the chronic gouty arthritis

Answer) Thanks for your valuable comment. the statement is revised as follows, “A fully human monoclonal antibody directed against RANKL, denosumab has been developed as a therapeutic option for bone loss in diverse pathological conditions such as metastatic lytic bone lesions, bone erosion in inflammatory arthritis, or low bone mineral density in osteoporosis [5]. Despite advances in the treatment of bone loss diseases, efforts on the identification of potent therapeutic agents with anti-osteoclast effect in gouty arthritis are still ongoing.”.

Newly added reference) 5. Miller, P.D. Denosumab: anti-RANKL antibody. Curr. Osteoporos. Rep. 2009, 7, 18-22.

  1. The paper has totally no literature on CKD-WID. It would be useful to describe CKD-WID and cite some studies using it. Moreover, it is not clear why CKD-WID was chosen over other specific HDAC6 inhibitors. Authors are recommended to enanched these points.

Answer) Thanks for your comment. We are sorry about not being able to answer your question clearly. Chong Kun Dang Pharmaceutical Corp., Seoul, Korea has developed diverse HDAC inhibitors for degenerative or inflammatory diseases. In addition to CKD-506, CKD-WID is one of the HDAC6 selective inhibitors developed by Chong Kun Dang Pharmaceutical Corp.. As far as we know, there is no clinical and experimental data about CKD-WID for other degenerative or inflammatory diseases.

   Please check the sentences described below in the part of introduction about why we chose HDAC6 inhibitors, as follows “Considering the association between HDAC6 and bone lesions, some investigations on HDAC6 related to osteoclastogenesis have been performed. Two studies demonstrated that HDAC6 did not play a meaningful role in osteoclast differentiation and formation [14, 15]. In contrast, functional loss of HDAC6 led to efficiently downregulate osteoclast differentiation and osteoclastogenic potency in different experimental cell lines [10, 12, 16]. The role of HDAC6 in osteoclasts differentiation and formation remains to be clarified in other pathogenic conditions.”. In addition, we add the sentence why we chose CKD-WID among HDAC6 inhibitors, as follows “Among HDAC6 inhibitors, the mechanism by which CKD-WID regulates osteoclast formation and differentiation in various pathological diseases has not yet been elucidated.”.

  1. The discussion is quite confusing, difficult to read and appears to be overloaded in relation to the results obtained. Further, there is no a clear separation between results obtained and those from other studies. Moreover, a clear logic line of presentation is lacking. The result is that the scientific progress provided by this work do not reach the reader efficaciously. Thus, I recommend to review the writing of this chapter in particular I encourage to simplify the test by removing extra data not correlated with results (for example HSP90) and by better underlying the results obtained.

Answer) We thanks for your valuable comment. We are so sorry that the results of our study were not clearly highlighted and that the results and lack of relevance were listed in the discussion section. Some parts of the discussion redeem this incompleteness, as many as possible. As you mentioned, we deleted the parts of HSP90 at line 189 and at line 209.

Reviewer 2 Report

No details available about HDAC6 inhibitor CKD-WID

The statement "It suggests that HDAC6 is considered a therapeutic target in osteoclastogenesis in gout." is a misinterpretation as this was not tested in patients with gout as a RCT.

Author Response

Dear Editor

Manuscript ID: pharmaceuticals-2195802

Title: Histone deacetylase 6 inhibitor CKD-WID suppressed monosodium urate-induced osteoclast formation by blocking calcineurin-NFAT pathway in RAW 264.7 cells

   Thank for the editor and reviewers of the ‘Pharmaceuticals’ for reviewing our manuscript. We have made some corrections and clarifications in the revised manuscript according to the editor’s or reviewer's comments. You can find out tracing marks for changes in revised manuscript. The changes are summarized below:

No details available about HDAC6 inhibitor CKD-WID

The statement "It suggests that HDAC6 is considered a therapeutic target in osteoclastogenesis in gout." is a misinterpretation as this was not tested in patients with gout as a RCT.

Answer) Thanks for your valuable comment. according to your recommendation, we revise the sentence as follows, “It suggests that HDAC6 is considered a therapeutic target in uric acid-mediated osteoclastogenesis.

Reviewer 3 Report

This manuscript explored the inhibition effect of CKD-WID on histone deacetylase (HDAC6) that related to regulation of osteoclast differentiation and formation. By feeding varied concentration of CKD-WID to the Murine monocyte/macrophage cells stimulated with MSU and RANKL, several osteoclast-related genes and transcription factors found be attenuated. This study provides a new possible method in gout treatment, although there are some minor issues in the manuscript need to be addressed.

1. The previous studies have already showed several effective HDAC inhibitors, what's the differences between them and CKD-WID? Why did this study focus on CKD-WID? The author should clarify this in discussion.

2. The manuscript tested several osteoclast-related genes and transcription factors, author should provide more information about how these genes/factors relate to osteoclast in the discussion. The Figure 5C looks good but lacks explanation.

3. Line 110-112, how did this fusion index was calculated?

4. Line 135-137, only c-Fos was repressed by similar level by three inhibitors (cyclosporin A, FK506 and CKD-WID). The effects to other markers were not "equal" in Figure 4E. Author should be caution to make such statement. 

5. Can authors also provide the results of negative control without M or R simulation (M+R -) in Figure 3A&B?

6. What's full name of CKD-WID?

Author Response

Dear Editor

Manuscript ID: pharmaceuticals-2195802

Title: Histone deacetylase 6 inhibitor CKD-WID suppressed monosodium urate-induced osteoclast formation by blocking calcineurin-NFAT pathway in RAW 264.7 cells

   Thank for the editor and reviewers of the ‘Pharmaceuticals’ for reviewing our manuscript. We have made some corrections and clarifications in the revised manuscript according to the editor’s or reviewer's comments. You can find out tracing marks for changes in revised manuscript. The changes are summarized below:

This manuscript explored the inhibition effect of CKD-WID on histone deacetylase (HDAC6) that related to regulation of osteoclast differentiation and formation. By feeding varied concentration of CKD-WID to the Murine monocyte/macrophage cells stimulated with MSU and RANKL, several osteoclast-related genes and transcription factors found be attenuated. This study provides a new possible method in gout treatment, although there are some minor issues in the manuscript need to be addressed.

  1. The previous studies have already showed several effective HDAC inhibitors, what's the differences between them and CKD-WID? Why did this study focus on CKD-WID? The author should clarify this in discussion.

Answer) Thanks for your comment. Unfortunately, CKD-WID used in this study is still an experimental drug, and we do not know exactly how to compare its efficacy with other HDAC6 inhibitors. To the best of our knowledge, no experimental data using CKD-WID has been published. The pharmaceutical company that provided this drug may have relevant data, but we do not have access to such sensitive and non-public information.

We focused on an inhibitory efficacy of a selective HDAC6 inhibitor CKD-WID on uric acid-induced osteoclast formation. In the first paragraph of the discussion section, supplementing the purpose of this study, we add the sentence that there is no study on the activation of osteoclasts in gout yet, as follow “Whether HDAC6 is involved in or responsible for MSU-induced bone damage in gout had not been examined.”.

  1. The manuscript tested several osteoclast-related genes and transcription factors, author should provide more information about how these genes/factors relate to osteoclast in the discussion. The Figure 5C looks good but lacks explanation.

Answer) Thanks for your kind comment. We revise or add some sentence about genes and factors related with osteoclast differentiation in the part of discussion, as follows “Consistently, our results also confirmed that HDAC6 inhibition using a selective HDAC6 inhibitor CKD-WID suppressed osteoclast differentiation and formation by inhibition of osteoclastogenic genes such asd c-Fos, TRAP, cathepsin K, and carbonic anhydrase II in MSU-primed RAW 264.7 cells in the presence of RANKL at line 191 – 194.” and “Consistently, this study confirmed that HDAC6 inhibition using CKD-WID also showed an anti-osteoclastic effect through suppression of calcineurin and NFATc1 activation in downsteram in the RANK-RANKL pathway.” at line 214 – 216.

   In addition, figure legend for Figure C is more detailed described as follows, “ C) Stimulation with both MSU and RANKL activates osteoclast-related markers c-Fos, NFATc1, calcineurin, cathepsin K, TRAP, and carbonic anhydrase II in RAW 264.7 cells. CKD-WID effectively suppresses osteoclast differentiation and formation through downregulation of c-Fos, cathepsin K, TRAP, and carbonic anhydrase II in MSU crystal-induced inflammation. Blockage of calcineurin-NFAT pathway by CKD-WID leads to suppression of Blimp1 a transcriptional regressor of anti-osteoclastogenic factor such as IRF-8 in gouty inflammation.”.

  1. Line 110-112, how did this fusion index was calculated?

Answer) Thanks for your comment. The fusion index was described in the figure legend as the following formula “Fusion index (%) was determined by dividing the total number of nuclei in the giant cells by the total number of nuclei x 100”.

  1. Line 135-137, only c-Fos was repressed by similar level by three inhibitors (cyclosporin A, FK506 and CKD-WID). The effects to other markers were not "equal" in Figure 4E. Author should be caution to make such statement.

Answer) Thanks for your valuable comment. We are so sorry about some incorrect description. The sentence is revised as follows “Western blot and densitometry showed that levels of osteoclast-related markers c-Fos, NFATc1, and cathepsin K protein expression were attenuated by treatment with cyclosporin A, FK506, and CKD-WID (Figure 4E). On the other side, only CKD-WID suppressed TRAP and carbonic anhydrase II protein expression under stimulation with MSU crystals and RANKL.”

  1. Can authors also provide the results of negative control without M or R simulation (M+R -) in Figure 3A&B?

Answer) Thanks for your valuable comment. We add 3 TRAP data including NC, RANKL, and MSU and their related TRAP (+) MNC and fusion index at Figure 3 A and B.

  1. What's full name of CKD-WID?

Answer) Thanks for your comment. We are sorry about not being able to answer your question clearly. Chong Kun Dang Pharmaceutical Corp., Seoul, Korea has developed diverse HDAC inhibitors degenerative or inflammatory diseases. In addition to CKD-506, CKD-WID is one of the HDAC6 selective inhibitors developed by Chong Kun Dang Pharmaceutical Corp.. It is just an experimental drug for which the official full name has not yet been determined.

Round 2

Reviewer 2 Report

Nil